# Functional Expression of TRPV1 Ion Channel in the Canine Peripheral Blood Mononuclear Cells

**DOI:** 10.3390/ijms22063177

**Published:** 2021-03-20

**Authors:** Joanna K. Bujak, Daria Kosmala, Kinga Majchrzak-Kuligowska, Piotr Bednarczyk

**Affiliations:** 1Department of Physics and Biophysics, Institute of Biology, Warsaw University of Life Sciences, 02-776 Warsaw, Poland; piotr_bednarczyk@sggw.edu.pl; 2Department of Physiological Sciences, Institute of Veterinary Medicine, Warsaw University of Life Sciences, 02-776 Warsaw, Poland; kosmala.daria92@gmail.com (D.K.); kinga_majchrzak@sggw.edu.pl (K.M.-K.)

**Keywords:** TRP ion channels, capsaicin, immune cells, PBMC

## Abstract

TRPV1, known as a capsaicin receptor, is the best-described transient receptor potential (TRP) ion channel. Recently, it was shown to be expressed by non-excitable cells such as lymphocytes. However, the data regarding the functional expression of the TRPV1 channel in the immune cells are often contradictory. In the present study, we performed a phylogenetical analysis of the canine TRP ion channels, we assessed the expression of TRPV1 in the canine peripheral blood mononuclear cells (PBMC) by qPCR and Western blot, and we determined the functionality of TRPV1 by whole-cell patch-clamp recordings and calcium assay. We found high expression of TRPV2, -M2, and -M7 in the canine PBMCs, while expression of TRPV1, -V4 and, -M5 was relatively low. We confirmed that TRPV1 is expressed on the protein level in the PBMC and it localizes in the plasma membrane. The whole-cell patch-clamp recording revealed that capsaicin application caused a significant increase in the current density. Similarly, the results from the calcium assay show a dose-dependent increase in intracellular calcium level in the presence of capsaicin that was partially abolished by capsazepine. Our study confirms the expression of TRPV1 ion channel on both mRNA and protein levels in the canine PBMC and indicates that the ion channel is functional.

## 1. Introduction

Ion channels are membrane proteins that regulate the transport of ions and membrane potential. Ion channels are also involved in many other cellular processes, such as the regulation of pH and volume, proliferation, motion, secretion, but most importantly they are mediators of cell–environment interactions, taking part in cell signaling cascades (i.e., calcium signaling) [1]. Dysfunctions of ion channels, their aberrant expression, or gene mutation often lead to many severe diseases. Channelopathies are responsible for such disorders as episodic ataxia, hyperkalemic and hypokalemic periodic paralysis, long and short QT syndrome, cystic fibrosis, neonatal diabetes mellitus, autosomal-dominant polycystic kidney disease, and hypomagnesemia with secondary hypocalcemia [2]. For this reason, ion channels are one of the most important molecular drug targets. After G-protein-coupled receptors (GPCRs), ion channels are the second-largest membrane protein group, and are targeted by around 18% of small molecule drugs according to the ChEMBL list [3,4]. Furthermore, recent studies pointed towards ion channels as being useful in the remote control of the cells. To date, at least four different ion channels, namely L-type voltage-gated calcium channel (Cav1.2), light-activated Cl^−^-conducting step-waveform inhibitory channelrhodopsin (SwiChR), transient receptor potential vanilloid 1 (TRPV1), and mechanosensitive channel Piezo1, have been utilized to control cell function or gene expression by subsequently applying electrical, light, magnetic, or ultrasound stimulation [5,6,7,8,9]. In the context of immunotherapy, the idea of using ion channels for remote control of chimeric antigen receptor (CAR) expression by T cells transfused into patients’ bloodstream might be of great importance, especially if it can ensure spatiotemporal precision of the therapy, as was shown by the study of Pan et al. [9].

Currently, it is widely known that ion channels are implicated also in the immune cell response. They are responsible not only for the transport of essential ions such as Mg^2+^ and Zn^2+^, but they also affect gene expression, proliferation, differentiation, cytokine secretion, cell migration, and apoptosis [10,11,12]. The network of ion channels ensures the proper intracellular oscillation of the calcium ions concentration, which plays a key role in the activation of lymphocytes [13]. Furthermore, dysfunction of ion channels is associated with impaired immune response. For instance, a mutation in stim1 and orai1 genes, encoding proteins of calcium release-activated channel (CRAC), causes immunodeficiency, while Kv1.3 channel block negatively influenced activation, proliferation, and interleukin-2 (IL-2) production by T cells [14,15]. Further studies also revealed that members of the transient receptor potential (TRP) ion channel family are among the channels expressed by the immune cells [16,17].

TRP ion channels are cation channels and polymodal receptors. They can be activated by a plethora of stimuli, such as temperature, pH, osmotic stress, reactive oxygen species (ROS), light, and endogenous and exogenous chemical compounds such as anandamide, LPS, and phytochemicals and flavonoids (capsaicin, eugenol, menthol, quercetin, etc.) [18,19,20,21,22,23,24]. To date, the expression of several TRP ion channels on the transcript level in the immune cells has been documented [12,25,26]. Ion channels from both melastatin (TRPM) and vanilloid (TRPV) subfamilies can be activated by temperature, H^+^, ROS, or hypoxia. Thus, they might take part in the regulation of immune response by these factors. Indeed, TRPV2 was documented as being expressed by the cells of both the innate and adaptive immune system, including macrophages, granulocytes, and lymphocytes [27]. Data regarding TRPV4 expression in immune cells are limited, however, as few studies associated these ion channels with the innate immune system, mostly with macrophages [28,29]. Nevertheless, a few studies showed protein expression of TRPV4 in the human T cells [30,31]. ROS-sensitive TRPM2 was, in turn, shown to be involved in T cell proliferation, cytokine secretion, and protection from oxidative stress-induced cell death, but also is associated with macrophage polarization [32,33]. TRPM5 and TRPM7 in turn were recently associated with the regulation of B cells and T cells functions, respectively [34,35]. The best-known member of the TRP ion channel family is TRPV1. It is also known as a receptor of capsaicin—a pungent compound from chili peppers [36]. In addition, TRPV1 is also activated by temperature (~42 °C), H^+^ ions, and lipopolysaccharide (LPS) [22,37,38]. Most importantly, TRPV1 was also documented to be expressed by the immune cells [30,39,40]. For instance, Bertin et al. [39] showed that TRPV1 colocalizes with immunological synapse and is involved in the regulation of CD4^+^ cell activation and functioning, shaping its proinflammatory properties. Involvement in the calcium signaling cascade, perception of stimuli from the environment, and finally expression of the TRPV1 ion channels in the immune cells suggest that it might participate in the regulation of the immune cell response. Nevertheless, the exact role of TRPV1 and other TRP ion channels, in general, is still elusive and not fully understood. This is partially due to the limited knowledge about these ion channels in the immune cells and conflicting results. For instance, Inada et al. [41] did not find expression of the *Trpv1* in both T and B cells in C57BL/6 male mice. On the contrary, Bertin et al. [39] documented the highest expression of *Trpv1* gene among other *Trpv* genes and confirmed that TRPV1 was functional using the same mouse strain (wild-type C57BL/6). Furthermore, in many studies regarding TRPV1 functional expression in the immune cells, capsaicin, a TRPV1 agonist, was used in high concentrations, ranging from 10 to 300 µM [42]. In contrast, in the studies on the nerve cells and transfected human embryonic kidney (HEK) cells, capsaicin in the range of nanomolar concentration was sufficient to evoke strong activation of the channel. These discrepancies need further refinement, especially in the context of differences arising from species and cell type. In the present study, we used canine primary peripheral blood mononuclear cells (PBMC), which are so-called non-excitable cells. Unlike in neurons or muscle cells, ion channels in non-excitable cells are not involved in generation action potential but rather are associated with regulatory functions [43]. Furthermore, early studies on the canine TRPV1, heterologously expressed in the HEK 293 cells, revealed that canine TRPV1 might be less sensitive to capsaicin, than other TRPV1 orthologs, due to a lack of conserved protein kinase A (PKA) phosphorylation site at position 117 (lysine instead of serine), responsible for capsaicin-sensitivity [44].

Our main goal was to determine the expression and functionality of the TRPV1 ion channel in peripheral blood mononuclear cells (PBMC) of the domestic dog (*Canis lupus familiaris*). Recent studies underlined the usefulness and value of the canine model in studies related to cancer biology and cancer drug discovery [45]. Dogs develop cancer spontaneously with a similar disease course to humans, and the treatment responses are also similar to humans. To date, there are more than 350 diseases of which the pathophysiology is comparable to humans [46]. Furthermore, the dog immune system also exhibits high similarity to that of humans, which supports its suitability for immune-oncology research [47]. However, to our knowledge, there are no current data regarding the functional expression of TRPV1 in the canine PBMC. In the present study, we included phylogenetics analysis of the canine TRP ion channels. We also evaluated the expression level of selected TRPV (vanilloid) and TRPM (melastatin) genes in the canine PBMC, with a special focus on TRPV1. Using patch-clamp recordings technique and calcium assay, we addressed the question about the functional expression of TRPV1 in the canine immune cells. The present study indicates for the first time that canine PBMC express the functional TRPV1 channel.

## 2. Results

### 2.1. Phylogenetic Analysis of Canine TRP Ion Channels Family and TRPV1 Orthologs

According to the available annotated sequences in the databases, domestic dogs potentially exhibit expression of 28 members of the TRP ion channels family. Similarly to humans and other vertebrates, canine TRP ion channels can be divided into six subfamilies, namely: TRPC (canonical), TRPM (melastatins), TRPV (vanilloids), TRPA (ankyrin), TRPML (mucolipins), and TRPP (polycystins) (Figure 1). The tree topology obtained using Bayesian analysis was similar to results obtained using maximum likelihood method.

The subfamily of canonical TRP ion channels in dogs is most probably represented by seven members that can be divided into three groups: (1) TRPC1, -C4, and -C5, (2) TRPC3, -C6, -C7, and (3) TRPC2. TRPC2 diverged from the other members of the TRPC subfamily. Based on the obtained phylogenetic trees, it can be indicated that the TRPV (vanilloids) group is divided into two subgroups: the first one includes TRPV1, -V2, -V3, and -V4, and the second one includes TRPV5 and -V6. Both TRPV5 and TRPV6 are not considered as thermoreceptors, in contrast to the four other members of the vanilloid subfamily, and exhibit high Ca^2+^ selectivity over other cations [48,49]. Canine TRPV1 is most closely related to the TRPV2, and together they form a sister group to TRPV4.

The amino acid sequence of the canine TRPV1 was compared to sequences from several distinct species: human, rhesus, mouse, rat, cat, rabbit, chicken, frog, and two species of teleost fish, namely zebrafish and tilapia. The analysis of the taxon relationship based on the protein sequences supports order of carnivora (dog and cat) as phylogenetically more closely related to primates (human) than rodents (Figure 2a). Thus, the dog model might have great potential for medical studies and potentially might help to circumvent the physiological limitations arising from murine models.

It is also well known that evolutionary changes in the TRPV1 sequences might influence the functionality of the ion channel, for instance sensitivity to its activators such as capsaicin. According to the literature, there are several residues responsive for capsaicin sensitivity of TRPV1, namely: Y511, S512, L518, M547, T550, or H570 [50,51,52]. In order to check capsaicin-sensitive aa positions, canine TRPV1 was compared to TRPV1 sequences from species-sensitive and insensitive species. Sequence analysis of TRPV1 orthologs revealed that amino acid positions responsible for capsaicin sensitivity are almost the same in humans and dogs (Figure 2b). However, the difference occurs at L518 position where leucine is changed to valine and L547 is changed to methionine in the canine TRPV1 sequence. However, the same changes are present in the sequence of rat TRPV1, which is documented to be a capsaicin-sensitive species. Furthermore, in capsaicin-insensitive species (zebrafish, tilapia, chicken) and less capsaicin-sensitive species (rabbit, Xenopus) there are aa substitutions in the positions homologous to human E570, T550, and S512 (Figure 2b).

### 2.2. TRPV1 Expression in the Canine PBMC

To determine whether canine PBMC express transcript for TRPV1, the qPCR analysis was performed. Additionally, mRNA expression of selected TRP ion channels (TRPV2, TRPV4, TRPM2, TRPM5, and TRPM7) was also tested. All of the mentioned TRP ion channels were documented to be expressed in the immune cells. qPCR analysis revealed that canine PBMCs express the TRPV1. Furthermore, transcripts of other TRP ion channels were also detected. However, major differences in the transcript abundance of selected TRP were observed. TRPV1 expression was relatively low, similarly to the expression of TRPV4 and TRPM5. Expression of TRPV2, TRPM2, and TRPM7 was, in turn, significantly (*p* < 0.0001) higher than the expression of TRPV1, -V4, and -M5 (Figure 3a).

TRPV1 expression in the canine PBMC was also confirmed on the protein level. Western blots on canine PBMC lysate probed with polyclonal TRPV1 antibody revealed immunoreactive bands around 100 kDa, which might correspond to TRPV1. The predicted weight of canine TRPV1 is 95.25 kDa. Additional bands were seen at around 120 kDa, ~200, and 250 kDa. Additional low-molecular weight bands were also observed in some experiments. We used antibodies that recognize human, non-human primate, mouse, and rat TRPV1 sequences. For this reason, we used human PBMC lysate to test antibodies’ specificity. There was only one immunoreactive band for human PBMC lysate at around 100 kDa (Figure 3b). The corresponding bands from canine samples were incorporated into further analysis.

Additionally, Western blot of PBMC membrane and cytosolic fraction revealed immunoreactive band for canine TRPV1 in the membrane fraction. Several bands, including bands around 100 kDa, were detected in the cytosolic fraction. GAPDH immunoreactive bands were only detected in cytosolic fractions (Figure 3d).

To check whether activation of T cells with known mitogen concanavalin A (ConA) influences TRPV1 protein expression level, canine PBMC were incubated with ConA for 24 h. Comparison between unstimulated (NS) and ConA-treated PBMC did not show any statistical difference in TRPV1 protein level after 24 h of incubation (Figure 3c).

### 2.3. Patch-Clamp Recording of Canine PBMC in Response to Capsaicin Treatment

To determine whether TRPV1 in PBMC is functional, two approaches were assessed: whole-cell patch-clamp recording and calcium assay using fluorescence calcium indicator Fluo-4. To avoid calcium fluxes associated with cell activation, both analyses were performed on cells that were not stimulated with ConA.

Whole patch-clamp recordings were performed to determine whether the canine PBMC are responsive to capsaicin, a potent TRPV1 channel agonist. Application of 16 µM capsaicin (CAP) significantly induced small inward currents (0.99 ± 0.13 pA/pF, *n* = 6) in comparison to control (0.49 ± 0.07 pA/pF, *n* = 6), when cells were held at -80 mV potential (Figure 4a). Such small current values differ significantly from what is known from studies on TRPV1 in the nerve cells or transfected HEK cells, where the current values reach the range of nA or µA. However, our results are in line with studies where current recordings were performed for the immune cells.

The current density was reduced in the presence of 20 µM capsazepine (CPZ) (0.55 ± 0.11 pA/pF, *n* = 6), reaching values similar to the values of the non-treated control (Figure 4b). These results point out that canine PBMC express ion channels that are sensitive to CAP treatment.

### 2.4. Effect of Temperature on Calcium Influx in Canine PBMC

To further verify the TRPV1 functionality in the canine PBMC, a calcium assay using fluorescent calcium indicator Fluo-4 was performed. Since TRPV1 is known to be activated by temperature around 42 °C, we measure the increase in fluorescence in response to increasing temperatures from 28 to 42 °C (Figure 5a).

We observed a significant increase in fluorescence intensity after short incubation at 42 °C in comparison to the control (28 °C), which might reflect activation of thermo-sensitive TRPV1 (Figure 5b). We also observed an increase in fluorescence in lower temperatures such as 35 or 38 °C when compared to control (28 °C) (Figure 5a). This phenomenon might be partially explained by the presence of other thermosensitive ion channels for instance TRPV4. Interestingly, we did observe a sharp increase in fluorescence (in the form of a spike) just after the incubation period. Additionally, we noticed that the response was dependent on the donor (cells from the donor 1 appeared to be less responsive to temperature incubation).

Furthermore, temperature alone did not lead to maximal fluorescence intensity, which was confirmed by using 10 ng/mL ionomycin as a positive control (Figure 5a,b).

### 2.5. Regulation of Intracellular Ca^2+^ Level by Capsaicin, Capsazepine and BAPTA

The intracellular Ca^2+^ level in the presence of the capsaicin, capsazepine, and 1,2-bis(o-aminophenoxy)ethane-N,N,N′,N′-tetraacetic acid (BAPTA) was also studied. We observed an increase in fluorescence after the application of capsaicin. The response to capsaicin was dose-dependent (Figure 6a).

Fluorescence intensity of the cells treated with 30, 50, and 100 µM was significantly higher in comparison to the control (Figure 6c). No effect was observed when lower doses of capsaicin such as 10 µM were applied (the data analyzed were taken from the first 2 min after capsaicin treatment). A high dose of capsaicin did not elicit the maximal level of fluorescence, which was proved by the addition of ionomycin (Figure 6b). Additionally, relative fluorescence raised significantly after ionomycin treatment.

For further analysis, we choose the lowest capsaicin concertation (30 µM) at which we observed the effect. To find out whether the increase in intracellular calcium concentrations is associated with TRPV1 activation, we used capsazepine, a known TRPV1 antagonist, and BAPTA, a calcium ion chelator. In the group pretreated with CPZ (30 µM), we observed lower fluorescence intensity after CAP treatment (Figure 7a).

Nevertheless, statistical analysis did not reveal significant differences between the groups CPZ + CAP and CAP alone in both the first two minutes (Figure 7c) and the total 10 min (Figure 7d) after CAP treatment. The results indicate that CAP-induced calcium influx was only partially inhibited by CPZ. Therefore, CPZ alone did not alter the basal fluorescence in comparison to non-treated cells.

Furthermore, BAPTA treatment was associated with a slow decrease in the cell fluorescence over time. Additionally, CAP treatment did not cause a rise in fluorescence intensity in the group pre-treated with BAPTA, which suggest that upon CAP-treatment calcium enters the cell from extracellular stores. We did not observe any effect of the vehicle (Dimethyl Sulfoxide (DMSO)) on the fluorescence intensity.

## 3. Discussion

TRP ion channels, including TRPV1, were mostly studied in the context of the sensory nervous system; however, increasing evidence indicates that they are also expressed by so-called non-excitable cells. Nevertheless, the data regarding functional expression of TRP ion channels in the immune cells are limited and the results are often contradictory. Here, we demonstrated that TRPV1 is expressed in the canine PBMC on both mRNA and protein levels. Additionally, the expression of the TRPV1 is confirmed by functional studies including patch-clamp and calcium assay.

TRP channels in mammals are divided into six subfamilies [20]. In humans, one member of the TRP channels family, namely TRPC2, is described as a pseudogene [53,54]. Our phylogenetic analysis showed that canine TRPC2 diverges from other TRPC channels. However, whether a domestic dog expresses functional TRPC2 is to be determined. Based on the phylogenetic analysis, TRPV1 is most closely related to TRPV2, which is in line with other phylogenetic analysis conducted on sequences from human, mice, and rat [55,56]. However, despite their evolutionary relationship, TRPV2, unlike TRPV1, is not sensitive to vanilloids [57].

Although TRPV1 was initially identified as a capsaicin receptor, further studies revealed that it is not sensitive to capsaicin across all species [36,58]. Gavva et al. [52] reported that rabbit TRPV1 is 100 times less sensitive to capsaicin than human and rat orthologs. Furthermore, other species such as chicken or fish are considered non-sensitive to capsaicin [58,59]. According to the molecular and electrophysiological studies, species-specific capsaicin-sensitivity was associated with a slight change in the specific amino acid residues of TRPV1 [50,51,52]. Based on the available literature data, several aa positions can be associated with the capsaicin sensitivity, including Y511, S512, L518, M547, T550, or H570 [50,51,52]. Most of these positions are the same in dogs and humans. However, we found differences in two positions—L518 (V) and L547 (M)—between human and dogs (canine amino acids are in parenthesis). Importantly, V518 and M547 are also present in the rat TRPV1, which is known to respond to capsaicin. In the study by Phelps et al. [44], the dog TRPV1 ortholog was cloned and stably transfected into HEK293 cell. The authors revealed that dog TRPV1 was sensitive to capsaicin; however, EC50 for canine TRPV1 was higher than for the rat, mouse, and human orthologs, and this was due to phenotypical differences.

We showed that canine PBMC express TRPV1; however, its expression on mRNA level is relatively low. There was no considerable difference in expression level between TRPV1, -V4, and -M5. Significantly, higher expression of TRPV2 than TRPV1 is in line with the data by Saunders et al. [40] regarding the expression of TRPV1 and TRPV2 in human PBMC. The authors described that the TRPV2 expression level was around 150-fold higher than TRPV1. Similarly, Spinsanti et al. [60] confirmed the highest expression of TRPV2 and the lowest expression of TRPV3 among TRPV1– 4 in human leukocytes. These data are in accordance with the results of our study on the canine PBMC. Nevertheless, Bertin et al. [39] demonstrated the highest mRNA expression of TRPV1 in comparison to other members of the vanilloid family and TRPM4 and TRPC3 in CD4^+^ lymphocytes from C57BL/6 WT mouse. On the contrary, Inada et al. [41] did not show expression of TRPV1 in mouse B and T cells. Such discrepancies in results may be due to inter- or intraspecies differences. The main rationale for choosing TRPV1 for further studies, despite its relatively low mRNA expression, was the availability of a specific activator (capsaicin) or antagonist. Furthermore, TRPV1 is documented to be activated by temperature, H^+^, and LPS—factors that might influence the function of the immune response.

The molecular weight of the TRPV1 channel is predicted to be around 95–100 kDa, which is in accordance with multiple studies. Interestingly, in some studies, additional bands for TRPV1 were also present, which might be associated with a glycosylated and non-glycosylated form of TRPV1. Bertin et al. [39] showed, using immunoblot analysis of murine splenic CD4^+^ lysate, two immunoreactive bands: one around ~95 kDa and the second around 115 kDa, most probably associated with glycosylation of TRPV1. Additionally, Veldhuis et al. [61] addressed TRPV1 glycosylation, demonstrating that it is involved in ionic permeability and desensitization of TRPV1. On the other hand, additional bands might also correspond to splice variants of TRPV1. TRPV1 was documented to have at least four splice variants: TRPV1a, TRPV1b, TRPV1var and short VR.5’sv [62,63]. In the present study, we also observed additional bands immunoreactive for TRPV1. However, whether the additional bands are glycosylated forms or splice variants of canine TRPV1 needs further refinement. Heavy bands around 150 and 250 kDa might correspond to multimers of TRPV1; however, this can only be speculated, since we used denaturing conditions. On the other hand, lower molecular weight bands were also observed occasionally. It is important to underline that in the present study, polyclonal anti-TRPV1 antibodies were used and thus we cannot exclude unspecific or cross-reactive binding of the antibodies.

Our immunoblotting analysis of cytosolic and membranous fractions showed a clear band for TRPV1 in the membranous fraction, which is in line with many studies that showed that TRPV1 predominantly occupies the plasma membrane, including immune cells [30,39,64]. However, in the cytosolic fractions, TRPV1 immunoreactive bands were also present. TRPV1 except for plasma membrane was shown to localize in the endoplasmic reticulum, Golgi apparatus, or mitochondria [65,66]. Nevertheless, several studies also documented TRPV1 distribution in the cytosol [66,67,68,69]. There are limited data directly addressing the cytosolic localization of TRPV1. More detailed studies need to be performed to answer the question of why TRPV1 gives a positive signal from cytosolic fraction.

Experimental evidence suggests that TRPV1 protein expression increases upon T cell activation. Majhi et al. [30] demonstrated that T cells (Jurkat, murine splenic T cells, and human T cell from peripheral blood) activated with either ConA or with CD3/CD28 exhibit upregulation of TRPV1 protein expression. Similarly, Ghoneum et al. [31] showed that TRPV1 expression increases in CD4^+^ T cells upon stimulation with CD3/CD28. In our studies, we did not found a significant increase in TRPV1 protein level in the cells treated with ConA for 24 h in comparison to the non-stimulated control. It is important to notice that the differences might arise depending on whether the studied immune cell population was homogenous (for instance pure CD4^+^ subpopulation) or, as in our case, heterogeneous.

Functional studies regarding the TRPV1 channel are mostly carried out using neurons and transfected HEK293 cells. Importantly, in these studies, the current values were around 1–5 nA at −80mV potential [70,71,72]. In some studies, the current values even reached a few microamps [73]. However, in our study, the values were definitely smaller, in the range of picoamps. Our data are in line with other electrophysiological studies on the immune cells, where the current values for TRPV1 were also small, at a few picoamps [39,74]. Small currents in the immune cells upon capsaicin treatment might arise from the low expression of the TRPV1. Our study showed that mRNA expression of TRPV1 is indeed significantly smaller than, for instance, TRPV2 in the PBMC. We also observed that not all of the cells exhibited sensitivity to capsaicin. One of the explanations is that PBMC consist of several distinct cell subpopulations, including B cell and T cells. Thus, we cannot exclude differences in TRPV1 expression, expression of different splice variants, or distinct patterns of post-translational modification in these distinct immune cell subpopulations and subsequent differences in TRPV1 activation. For instance, Lu et al. [75] identified a functional TRPV1 splice variant named TRPV1b, which is activated by temperature but not by capsaicin or protons. Similarly, Vos et al. [63] demonstrated that TRPV1b can negatively regulate TRPV1 functions including responsiveness to capsaicin. Interestingly, studies by Shin et al. [76] showed that TRPV1 heterologously expressed in the HEK cells had greater sensitivity to CAP and its synthetic analog than native TRPV1 from sensory neurons while neurons appeared to be more sensitive to resiniferatoxin. The authors suggested that the difference might arise either from the subtypes of TRPV1 ion channel or some regulatory mechanisms that are different in HEK in comparison to sensory neuron cells. Currently, it is well known that activation of the TRPV1 channel is modulated by post-translational modifications such as phosphorylation or glycosylation [77]. Such modifications of the TRPV1 channel affect its activity, ligand binding, or even cell membrane localization [53,65]. Furthermore, TRPV1 sensitivity to capsaicin appeared to be tuned by phosphatidylinositol-4,5-bisphosphate (PIP2). Prescott et al. [78] showed that mutations of sites involved in PIP2–TRPV1 interaction were associated with a lower activation threshold for chemical and thermal stimuli. Lastly, the possibility of heterotetramers formation by TRPV1 with TRP channels, for instance, TRPA1, or functional coupling with other ion channels such as BK(mSlo1), might also have a great influence on TRPV1 ion activity [70,79]. Hence, in different cell types, TRPV1 can exhibit distinct activation modes and thresholds.

Some of the TRP ion channels, including TRPV1, -V2, -V3, -V4, and TRPM2, exhibit sensitivity to temperature. TRPV1 is well documented to have an activation threshold around 42 °C while TRPV4 is activated around 35 °C [80,81]. Additionally, TRPM2 potentially can be activated by a temperature around 35 °C [82]. In our study, we showed that intracellular calcium concentration was increasing along with the rise in temperature from 28 to 42 °C. Additionally, after every incubation period, we observed a sharp increase in fluorescence levels. We assumed that the observed spikes are associated with the activation of ion channels; however, this needs further clarification. Statistical analysis revealed that an increase in fluorescence intensity was statistically significant at 42 °C (data from the first measurement) in comparison to control. A further increase was observed upon repeated incubation at 42 °C. An increase in calcium levels in response to temperature clearly emphasizes the participation of ion channels; however, more detailed studies are required to further identify which channels are involved. The results suggest that thermo-sensitive TRP ion channels might partially be responsible for the calcium influx in response to temperature. Our findings regarding the expression of TRPV1 and other thermosensitive TRP channels such as TRPV4 and TRPM2 further support the assumption about TRP channels involvement in the thermal response. On the other hand, Xiao et al. [83] underlined the role of the STIM1/Orai1 complex in temperature sensing by the immune cells. The authors showed that high-temperature blocks ICRAC currents and calcium influx associated with the functional coupling of STIM1/Orai1 followed by Ca^2+^ influx as a response after cooling. Temperature-dependent activation of ion channels might have a profound effect on cell activity. Thus, understanding the molecular mechanisms that govern temperature—ion channel—cell physiology interactions might help in immune cell modulation.

Capsaicin, a pungent compound from chili peppers, is a well-known TRPV1 agonist. In the present study, CAP treatment elicits an increase in fluorescence values with a dose-dependent manner, which is in accordance with multiples scientific reports. For instance, Mamatova et al. [84] documented that in the HEK293 cells transfected with rat TRPV1 that capsaicin-induced currents were dependent on capsaicin concentration (range from 10 to 50 µM). Surprisingly, we did not observe any increase in fluorescence with lower capsaicin concentration, such as 10 µM. In most of the studies, capsaicin evokes a strong current even at nanomolar concentrations. In the study by Phelps et al., authors for the first time expressed in the HEK293 cells and functionally characterized canine TRPV1. The authors confirmed the dose-dependent responsiveness of TRPV1 and showed that CAP at the concentration of 0.03 µM elicited maximal signal [44]. Wang et al. [85] showed dose–response of heterologous expressed murine TRPV1 with 0.01 µM concentration as the lowest tested concentration able to stimulate TRPV1-dependent calcium influx. However, Wang et al. [86] found that CAP did not elicit calcium influx at concentrations lower than 100 µM in the neutrophils. It can be speculated that the level of capsaicin response is somehow dependent on the cell type. Immune cells are so-called non-excitable cells and most probably their ion channels-driven response might be more subtle [87]. Hence, it is difficult to compare the magnitude of response between the transfected cells where particular ion channels are overexpressed and non-excitable cells such as lymphocytes where the channel expression in many cases is less abundant. Similar assumptions were made by Pecze et al. [88]. The authors shed light on the fact that neurons are more sensitive to capsaicin than cancer cells with confirmed TRPV1 expression. However, overexpression of TRPV1 channels in the cancer cells made them more sensitive to capsaicin, indicating that TRPV1 expression level is associated with the magnitude of the response [88]. Interestingly, upon CAP application, the kinetics of calcium influx was slower than expected. One of the potential explanations is heterotetramerization. In the studies by Cheng et al. [89], the authors showed that for heteromeric TRPV1/TRPV3 channel CAP, the dose–response curves had shallower slopes than homotetramer of V1. Interestingly, in the same study, the authors also used a high concentration of CAP (10 and 30 µM) [89].

Capsazepine is known competitive antagonist of capsaicin. However, in the present study, 30 µM CPZ failed to inhibit completely increase in fluorescence upon addition of 30 µM CAP. Pearce et al. [90] noted that observed agonist (CAP) action can be observed even in the presence of antagonists (for instance CPZ) due to differential rates of cell penetration. The authors showed that 30 times higher concentration of CPZ blocked almost completely the response to CAP, while lower concentrations led to a smaller reduction in intracellular calcium influxes caused by an agonist of TRPV1. Thus, the increase in fluorescence intensity after capsaicin treatment can be observed and is dependent on the antagonist–agonist ratio. Interestingly, CPZ was shown previously to cause an increase in calcium level in neutrophils at a high micromolar concentration (>100 µM) [86]. Additionally, CPZ was shown to cause an increase in calcium levels in neurons what was associated with activation of TRPA1 [91]. Additionally, increasing evidence indicates that TRPV1 and TRPA1 colocalize and may form functional heterotetramers characterized by distinct electrophysiological properties [71,92]. Recent studies reported that TRPA1 is expressed by the T lymphocytes [93,94]. Moreover, TRPA1 was reported to inhibit activation of TRPV1 [94]. Complex interactions between these and other ion channels in the primary cells might thus obscure the results.

In order to verify that the increase in fluorescence is driven by extracellular calcium influx, we preincubated the cells in the presence of Ca^2+^ ions chelator- BAPTA. We did not observe any increase in fluorescence in response to CAP in the groups pretreated with BAPTA, which indicates that the increase in fluorescence intensity after CAP administration was associated with Ca^2+^ influx from the extracellular stores. Surprisingly, we observed a decrease in fluorescence in the groups pretreated with BAPTA over time. Recently, Qiu et al. [95] showed that treatment of Fluo-8AM-loaded protoplast of *Malus domestica* with EGTA also reduced the calcium fluorescence intensity. Additionally, Palmer et al. [96] noted that the addition of EGTA or BAPTA leads to a slow decline in fluorescence intensity, which supports our observation.

## 4. Materials and Methods

### 4.1. Phylogenetic Analysis

Canine TRP ion channels protein sequences were collected from the Protein database available from National Center for Biotechnology Information (https://www.ncbi.nlm.nih.gov/protein/; accessed on 26 June 2019) and Ensembl Genome Browser (https://www.ensembl.org/index.html; accessed on 26 June 2019). The canine voltage-gated potassium channel (Kv1.4) sequence was used as an outgroup (sequences and accession numbers are in a Appendix A). Clustal Omega was used to perform multiple sequence alignment. The obtained alignment was refined manually and then trimmed automatically using G-block on multiplatform Seaview [97,98,99]. The analysis was performed using Mr. Bayes 3.2.6 (four Markov chains, 100,000 generations with the tree sampling frequency of 500 and burning fraction 25%) on the NGPhylogeny.fr platform and maximum likelihood method using ProML (bootstrapping was performed using 500 pseudo-replicates, JTT model was chosen as amino acid substitution model), and the tree was generated in Consense (majority rule was used as a consensus type) using PHYLIP package 3.695 [100,101,102,103]. The phylogenetic tree was formatted in FigTree (v 1.4.4.).

For the TRPV1 orthologs analysis, TRPV1 protein sequences from the following species—dog (*Canis lupus familiaris*, Acc. No.: NP_001003970.1), human (*Homo sapiens*, Acc. No.: NP_061197.4), rhesus (*Macaca mulatta*, Acc. No.: XP_028691720.1), cat (*Felis catus*, Acc. No.: XP_003996439.1), rabbit (*Oryctolagus cuniculus*, Acc. No.: NP_001075635.1), mouse (*Mus musculus*, Acc. No.: NP_001001445.1), rat (*Rattus norvegicus*, Acc. No.: NP_114188.1), chicken (*Gallus gallus*, Acc. No.: XP_024997658.1), western clawed frog (*Xenopus tropicalis*, Acc. No.: XP_031751643.1), zebrafish (*Danio rerio*, Acc. No.: NP_001119871.1), and tilapia (*Oreochromis niloticus*, Acc. No.: XP_013120120.1)—were obtained from the NCBI Protein database. Sequences were aligned in Seaview using MUSCLE algorithm and refined manually. The sequences were then aligned with MAFFT, curated by BMGE and the tree was inferred using FastMe on https://ngphylogeny.fr platform (accessed on 26 June 2019) and formatted in Interactive Tree of Life available from https://itol.embl.de (accessed on 26 June 2019).

### 4.2. PBMC Isolation and Cell Culture

Peripheral blood mononuclear cells (PBMC) were isolated from the full blood of healthy dogs (donors of the veterinary blood bank). Isolation of PBMC was conducted within 4 h after blood collection. Isolation was carried out using density gradient medium Histopaque-1077 (Merck, St. Louis, MO, USA) and SepMate^TM^-50 tubes (StemCell Technologies, Vancouver, Canada) according to the StemCell technologies protocol. In case of PBMC contamination with erythrocytes, the cell pellet was treated with 1X Red Blood Cell Lysis Buffer (ThermoFisher Scientific, Waltham, MA, USA) according to the manufacturer’s instructions. Cells were counted and cryopreserved in FBS supplemented with 10% DMSO (Merck, St. Louis, MO, USA) for further use.

Upon thawing, cells were counted and their viability was determined using trypan blue dye and cell counter EVETM (NanoEnTek Inc., Seul, South Corea). Furthermore, after thawing, cells were left overnight to rest. PBMCs were cultured in the RPMI-1640 medium with GlutaMAX (ThermoFisher Scientific, Waltham, MA, USA) supplemented with MEM non-essential amino acids (Gibco^TM^, Thermofisher Scientific, Waltham, MA, USA), sodium pyruvate (Gibco^TM^, Thermofisher Scientific, Waltham, MA, USA), fetal bovine serum with final concentration 10% (Merck, St. Louis, MO, USA) and PenStrep (Gibco^TM^, Thermofisher Scientific, Waltham, MA, USA). Cells were kept at 38.5 °C, 5% CO_2_, and 95% humidity. Cells were incubated on the adherent plates to reduce the number of monocytes and subsequently enrich lymphocyte fraction (PBL).

For the purpose of the PBMC activation, cells were incubated with the natural plant mitogen, concanavalin A (ConA) (Merck, St. Louis, MO, USA), in a final concentration of 5 µg/mL. After 24 h, cell activation was evaluated visually with light microscopy Olympus CKX41 (Olympus Corporation, Tokyo, Japan) and flow cytometry based on the expression of CD25 marker (eBiosciences^TM^, Thermo Fisher Scientific, Waltham, MA, USA) (data not shown).

### 4.3. RT-qPCR

Total RNA was extracted using the RNAqueous^TM^- Micro Total RNA Isolation Kit (Thermo Fisher Scientific, Waltham, MA, USA). The isolated samples were treated with DNase A in order to avoid potential genomic DNA contamination. RNA concentration and A260/230 and A20/280 was determined on NanoDrop 2000 spectrophotometer (Thermo Fisher Scientific, Waltham, MA, USA). One microgram of RNA was converted to cDNA using High-Capacity RNA-to-cDNA^TM^ (Applied Biosysyems^TM^, Thermo Fisher Scientific, Waltham, MA, USA) according to manufacturer’s recommendations. Quantitative PCR reactions were performed in 96-well plates FrameStar^®^ (4TiTude, Brooks Life Sciences, Chelmsford, MA, USA) using SYBR^®^ Green PCR Master Mix (Applied Biosysyems^TM^, Thermo Fisher Scientific, Waltham, MA, USA) in Stratagene Mx3005P thermocycler (Agilent Technologies, Santa Clara, CA, USA). The following qPCR conditions were used: 2 min of initial denaturation at 95 °C, 40 cycles at 95 °C for 15 s and 58 °C for 15 s and 72 °C for 60 s, followed by dissociation curve analysis. Potential primer–dimer formation was checked in “no template control”, and the molecular size of the amplicons was confirmed by 2.5% agarose gel electrophoresis (Appendix A). Primers used in this study were designed de novo using either Primer Blast and Primer3 software [104,105]. Primers for normalization gene RPS-19 have been described previously [106] (Table 1). Data analysis was performed using the 2^−ΔCt^ method [107].

### 4.4. Western Blot

Harvested PBMCs were resuspended in the ice-cold RIPA buffer (Thermo Fisher Scientific, Waltham, MA, USA) supplemented with Halt^TM^ Protease Inhibitor Cocktail 100X (Thermo Fisher Scientific, Waltham, MA, USA). The cell lysate was centrifugated at 11,000× *g* for 15 min at 4 °C and the supernatant was transferred into fresh tubes. Protein concentration was determined using Pierce^TM^ BCA Protein Assay Kit (Thermo Fisher Scientific, Waltham, MA, USA), and absorbance was measured on the Tecan Infinite M200 Spectrophotometer (Tecan, Männedorf, Switzerland) at 562 nm wavelength. Twenty micrograms of proteins were mixed with NuPAGE^TM^ LDS Sample Buffer (Thermo Fisher Scientific, Waltham, MA, USA) supplemented with NuPAGE^TM^ Sample Reducing Agent (Thermo Fisher Scientific, Waltham, MA, USA). Samples were incubated at 70 °C for 10 min and loaded on 4–20% Mini-PROTEAN^®^ TGX^TM^ Precast Protein Gels (Bio-Rad laboratories, Hercules, CA, USA). As a protein weight marker, Precision Plus Protein^TM^ All Blue Prestained Protein Standard (Bio-Rad Laboratories, Hercules, CA, USA) was used. Electrophoresis was performed in Tris/glycine/SDS buffer (Merck, St. Louis, MO, USA) in the Mini-PROTEAN^®^ Tetra System (Bio-Rad laboratories, Hercules, CA, USA) for around 35-45 min at a constant voltage of 150 V. After electrophoresis, proteins were electrotransferred on a nitrocellulose membrane in Tris/glycine buffer for Western blots and Native Gels transfer buffer (Bio-Rad Laboratories, Hercules, CA, USA) with 20% methanol content for 70 min at constant 100V. After transfer, membranes were blocked in 5% non-fat milk in TBST buffer (Tris-buffered saline with Tween 20: 20 mM Tris, 140 mM NaCl and 0.1% Tween 20 (Merck, St. Louis, MO, USA)) for 60 min at room temperature. After blocking membranes were incubated overnight with primary antibodies: rabbit polyclonal anti-TRPV1 (1:1000, Thermo Fisher Scientific, Waltham, MA, USA) and mouse anti-β-Actin (1:1000, Santa Cruz Biotechnology Inc., Dallas, TX, USA) or mouse anti-GAPDH (1:1000, Thermo Fisher Scientific, Waltham, MA, USA) at 4 °C with constant agitation. Membranes were then washed 3 times for 10 min in TBST buffer and incubated with secondary antibodies IRDye^®^ 800CW and 680RD (LI-COR Biosciences, Lincoln, NE, USA) for around 45 min in the absence of light. Following 3 × 10 min washes in TBST in the dark, membranes were air-dried and scanned on Odyssey (LI-COR Biosciences, Lincoln, NE, USA). Band intensities were analyzed in the Image Studio Lite version 5.2 software (LI-COR Biosciences, Lincoln, NE, USA) and normalized to β-actin.

Membrane and cytosolic fraction were isolated using Mem-PER^TM^ Plus Membrane Protein Extraction Kit (Thermo Fisher Scientific, Waltham, MA, USA) according to the manufacturer’s recommendations. Halt^TM^ Protease Inhibitor Cocktail (Thermo Fisher Scientific, Waltham, MA, USA) was added to both the permeabilization and solubilization buffers. Centrifugations were performed at 4 °C.

### 4.5. Patch-Clamp

Whole-cell recordings were performed using the Axopatch 200B Microelectrode Amplifier (Molecular Devices, San Jose, CA, USA), the Digidata 1322A 16-bit data acquisition system (Molecular Devices, San Jose, CA, USA), and the inverted microscope Axiovert 10 (Carl Zeiss, Oberkochen, Germany). Microelectrodes were pulled from borosilicate glass (Harvard Apparatus, Holliston, MA, USA) on a horizontal micropipette puller P-97 (Sutter Instrument Company, Novato, CA, USA). When filled with solution, electrodes have around 6–9 MΩ resistance. The extracellular and intracellular solutions were prepared as previously described by Bertin et al. [39] (in mM, extracellular solution: 144 NaCl, 5 KCl, 2 MgCl_2_, 1 CaCl_2_, 10 glucose, 10 HEPES; intracellular solution: 126 K-gluconate, 10 KCl, 5 EGTA, 4 MgATP, 10 HEPES). Data were recorded in the whole-cell configuration, at 5 kHz sampling frequency at 35 °C. The measurements were performed using a voltage ramp from −80 mV to +80mV (20 mV steps) prior to and after compound addition. Capsaicin (Merck, St. Louis, MO, USA) in the final concentration of 16 µM and capsazepine (20 µM) (Merck, St. Louis, MO, USA) was diluted in the extracellular solution, and administrated using peristaltic pump Reglo (Ismatec, Wertheim, Germany). Data acquisition and analysis were performed using Clampex 10.7 software and Clampfit 10.7 software subsequently (Molecular Devices, San Jose, CA, USA).

### 4.6. Calcium Assay

Ca^2+^ flux measurements were performed using the Calcium Assay Kit (Thermo Fisher Scientific, Waltham, MA, USA). Cells were incubated with Fluo-4 Direct^TM^ fluorescent dye supplemented with 5 mM probenecid for 30 min at 37 °C. Cells were treated with capsazepine (CPZ) at the final concentration of 30 µM or with a chelator of calcium ions BAPTA (Merck, St. Louis, MO, USA) at the final concentration of 0.5 mM around 20 min prior to measurements. Changes in fluorescence intensity were measured on Fluoroskan Ascent FL Plate Reader (Thermo Fisher Scientific, Waltham, MA, USA) at 538 nm emission wavelength, with the kinetics interval set to 3 s, and at 28 °C. For each sample, measurements were performed for 4 technical replicates. Capsaicin (CAP) was perfused after the first 40 measurement cycles, in a concentration range of 10–100 µM. As a positive control, ionomycin in a final concentration of 1 ng/mL was used. Cells that were not treated with any drugs constituted a negative control. DMSO was used as a vehicle in the final concentration of 0.1%. In the experiment regarding temperature effect, fluorescence measurements were performed after a 3 min incubation period at a temperature range from 28 to 42 °C. The results were normalized to the basal fluorescence (F_0_) and presented as F/F_0_ ratio. The area under the curve (AUC) was calculated and values were used for statistical analysis of temperature and agonist treatment.

### 4.7. Statistics

Statistical analysis was performed using GraphPad Prism 5.01 software (GraphPad Software, San Diego, CA, USA) and the data were represented as mean + s.e.m. (bars) or median ± minimal and maximal values (box). Data were analyzed using Student’s t-test, one-way ANOVA, followed by Tukey’s or Dunnett’s (comparison to control) post hoc test or two-way ANOVA (indicated in Figures). ANOVA assumption of homogeneity of variances was tested by Bartlett’s test. Significance levels were denoted with asterisks: * *p* < 0.05, ** *p* < 0.01, *** *p* < 0.001.

## 5. Conclusions

To our knowledge, this is the first work that describes canine TRPV1 in the PBMC. In summary, canine TRPV1 is more closely related to human TRPV1 than the murine ortholog. Furthermore, we have shown that canine PBMC express TRPV1 ion channels on both the mRNA and protein level. Canine TRPV1 localizes in the plasma membrane of PBMC; however, other subcellular localization of the TRPV1 channel cannot be excluded. Furthermore, we showed that PBMC respond to capsaicin in a dose-dependent manner, but the response was only partially blocked by CPZ. The magnitudes of response upon capsaicin treatment were lower in the canine PBMC in comparison to the data from studies regarding neuronal cells or heterologously expressed TRPV1. A potential explanation is that PBMC exhibit rather low expression of the TRPV1 channel. Additionally, PBMC are non-excitable cells that are characterized by distinct physiology than excitable cells such as neurons. The difference between neurons and heterologous expression systems versus primary cells such as lymphocytes needs further refinement, especially in terms of agonist and antagonist doses and responses. Additionally, the likelihood of heterotetramers formation needs to be taken into consideration, especially in the context of their diverse functionality. Our study paves the way for further studies regarding canine TRP ion channels and their putative role in immune cell functions.

## Figures and Tables

**Figure 1 ijms-22-03177-f001:**
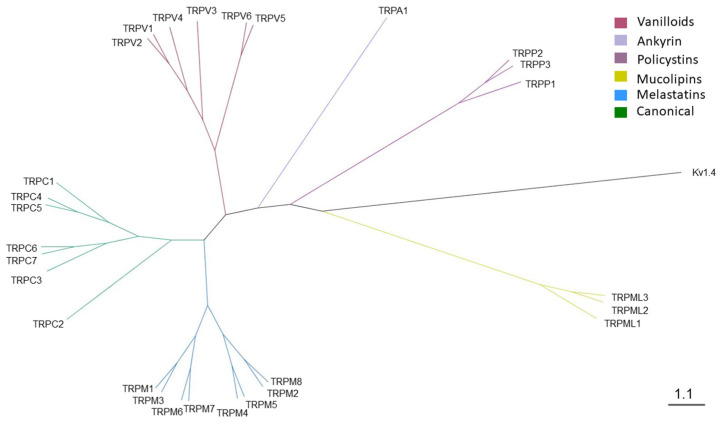
Phylogenetic tree of the canine transient receptor potential (TRP) ion channels family. Six different subfamilies, namely TRPC (canonicai), TRPM (melastatin), TRPV (vanilloid), TRPA (ankyrin), TRPP (policystin), and TRPML (mucolipin), were denoted in different colors. The tree was constructed using Bayesian analysis (four Markov chains, 100,000 generations, the tree sampling frequency of 500 and burning fraction 25%) of the available canine TRP ion channel protein sequences. Canine voltage-gated potassium channel Kv1.4 was used as an outgroup. Scale bar indicates branch length.

**Figure 2 ijms-22-03177-f002:**
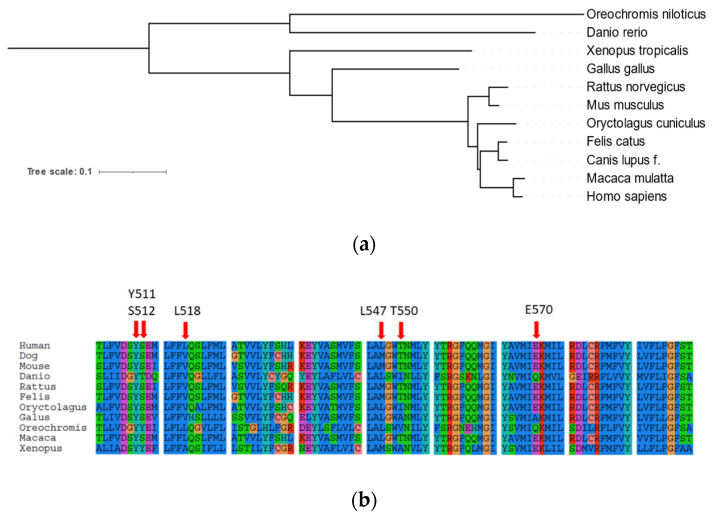
Analysis of TRPV1 orthologs. (**a**) Phylogenetic tree of TRPV1 orthologs. TRPV1 of Carnivora is more closely related to primates than to rodents. The tree was constructed using available TRPV1 sequences and the FastMe algorithm. The tree was visualized using Interactive Tree of Life. Scale Bar indicates branch length. (**b**) Comparison of TRPV1 orthologs amino acid positions known from the literature to be involved in capsaicin sensitivity. Canine TRPV1 sequence shares similarity to human, mouse, rat, and cat in terms of capsaicin-sensitive aa positions. Substitutions at these positions are associated with the channel that is less responsive or not responsive to CAP as in the case of rabbit, chicken, frog, and fish.

**Figure 3 ijms-22-03177-f003:**
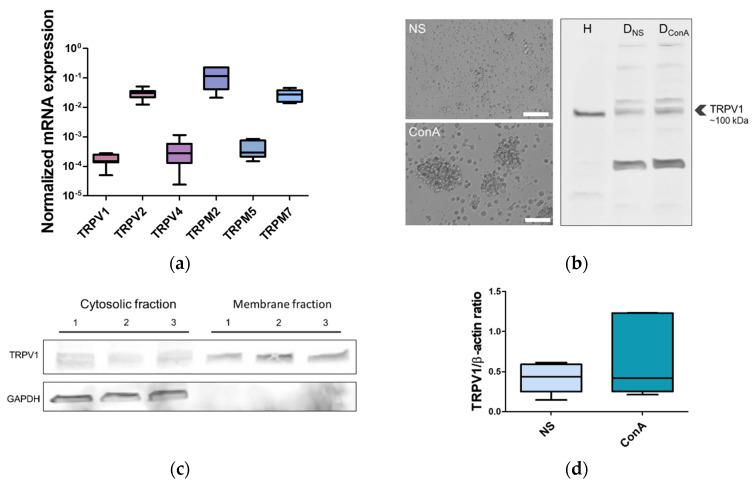
Expression of TRP ion channels in the canine PBMC, with special emphasis given to TRPV1. (**a**) mRNA expression of selected TRP ion channels in the canine PBMC. TRPV2, TRPM2, and TRPM7 were found to exhibit high expression, while TRPV1, TRPV4, and TRPM5 were characterized by low expression. Expression level is represented as arbitrary units. Data are shown as a dot plot, with mean ± s.e.m. (*n* = 7 animals). Different letters indicate significant differences (one-way ANOVA, followed by Tukey’s post hoc, *p* < 0.0001). (**b**) TRPV1 protein level in the canine PBMC. TRPV1 antibody recognized a major band around ~100 kDa in the lysate of the canine PBMC. As a control of antibody specificity, we used PBMC lysate from a human (H). Western blot was run on PBMC that were not stimulated (D_NS_) or activated with plant mitogen concanavalin-A (D_ConA_) for 24 h. Activation was confirmed by microscopy (activated cells groups into clusters). Scale bars, 50 µm. (**c**) Western blot’s densitometric quantification. We did not found significant differences between NS and ConA treated groups. Β-actin was used as a loading control. Values are means ± s.e.m., *n* = 5, Student’s *t*-test, *p* < 0.05. (**d**) Western blot of cytosolic and membranous fractions from canine PBMC. TRPV1 localizes in plasma membrane; however, immunoreactive bands in cytosolic fractions suggest that it might also localize in other subcellular compartments. As a control, we used GAPDH (~40 kDa). Immunoreactive bands for GAPDH were found only in the cytosolic fractions (*n* = 7).

**Figure 4 ijms-22-03177-f004:**
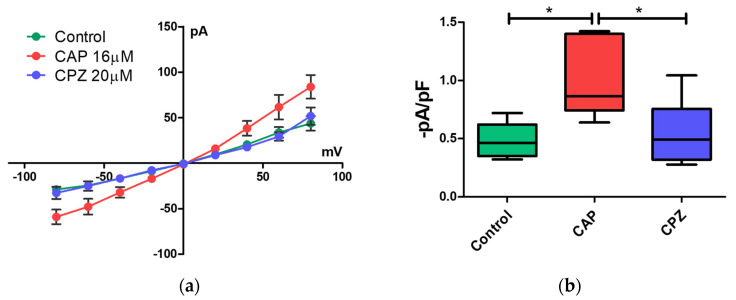
Whole-cell patch-clamp recordings. (**a**) Voltage-current plots before (control—green dots) and after application of 16 µM capsaicin (CAP—red dots) and 20 µM of capsazepine (CPZ—blue dots). Presence of CAP increased the ion currents. Subsequent perfusion with CPZ lower CAP-induced ion currents to a level similar to the control. (**b**) Current density of control, CAP-, and CPZ-perfused cells at −80mV. A significant increase in current density was observed in the CAP-treated cells in comparison to control and CPZ. Significance was denoted with an asterisk (*). Values expressed as mean ± s.e.m., *n* = 6, one-way ANOVA, with Tukey’s post hoc test (*p* < 0.05).

**Figure 5 ijms-22-03177-f005:**
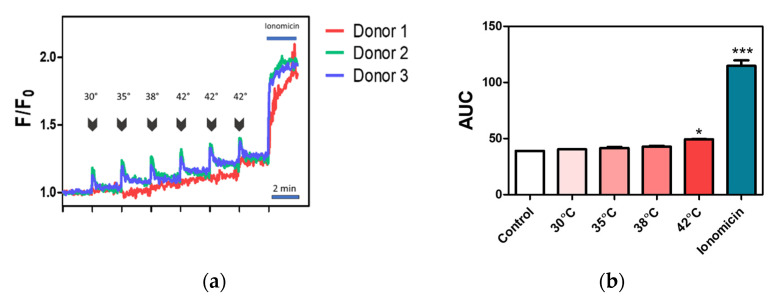
Temperature effect on intracellular calcium levels. (**a**) Relative calcium levels were measured using fluorescent calcium indicator Fluo-4. An increase in temperature was associated with raise in fluorescence intensity (associated with an increase in calcium levels). Ionomycin was used as a positive control. Calcium traces are presented as F/F0 ratio, *n* = 3. (**b**) Short (3 min) preincubation of cells at 42 °C temperature caused a significant increase in fluorescence intensity associated with raise in intracellular calcium levels. Ionomycin at final concertation of 1 ng/mL caused a further increase in intracellular calcium. Bars represent mean ± s.e.m., *n* = 3 of the area under the curve (AUC) of calcium traces. Significant differences were denoted with asterisks, depending on the p- level: * *p* < 0.05, *** *p* < 0.0001 (one-way ANOVA, followed by Dunnett’s post hoc test).

**Figure 6 ijms-22-03177-f006:**
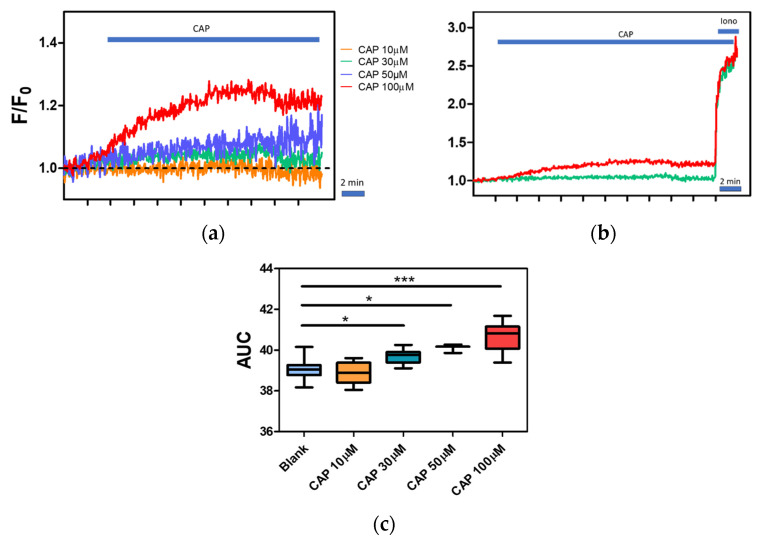
Capsaicin dose–response of the canine PBMC. (**a**) Calcium traces of the canine PBMC treated with different CAP doses. An increase in fluorescence intensity was observed with increasing agonist (CAP) concentration. (**b**) Administration of ionomycin in the final concentration of 1ng/mL led to a further increase in fluorescence intensity. (**c**) Significant increase in intracellular calcium levels was observed in groups treated with 30, 50, and 100 µM CAP concentrations when compared to control. No effect was observed in the group treated with 10 µM CAP. Data are shown as a dot plot with mean ± s.e.m of AUC (from first 2 min), *n* = 3–8. Significant differences were denoted with asterisks, depending on the *p*- level: * *p* < 0.05, *** *p* < 0.0001 (one way ANOVA, followed by Tukey’s post hoc).

**Figure 7 ijms-22-03177-f007:**
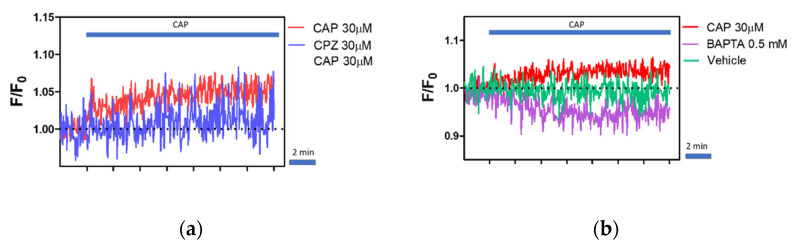
Capsazepine only partially inhibits CAP-induced calcium influx. (**a**) Calcium traces of cells treated with 30 µM CAP and 30 µM CPZ before CAP treatment. The presence of CPZ inhibits a slight increase in fluorescence intensity upon CAP treatment. Data are represented as the mean of the traces calculated as F/F0. (**b**) Calcium traces of cells pretreated with 0.5 mM BAPTA or DMSO as a vehicle. Administration of 30 µM CAP did not cause an increase in fluorescence in cells treated with BAPTA. No changes were observed in the vehicle group. Data are represented as the mean of the traces calculated as F/F0. (**c**,**d**) Area under the curve (AUC) of cells treated with CAP (30 µM) only, CPZ (30 µM) and CAP (30 µM), BAPTA (0.5mM) and CAP (30 µM), and treated only with vehicle (DMSO) instead of CAP. Bars represent mean ± s.e.m of AUC from the first 2 min (**c**) and total (10 min) (**d**) post CAP application. The solid line represents basal fluorescence. Data were analyzed by one-way ANOVA with Tukey’s post hoc test and two-way ANOVA, n = 3–7. Factorial effects were denoted as “Drugs”, “CAP”, and Interaction. Significance was denoted with asterisks (*), p- level: * *p* < 0.05,

**Table 1 ijms-22-03177-t001:** Primer sequences for real time PCR of selected canine TRP ion channels. RPS-19 was used as a normalization gene.

Target Name	Accession Number	Forward Primer	Reverse Primer	Size (bp)
TRPV1	NM_001003970.1	AACATGCTCATTGCCCTCAT	GACCTCATCCACCCTGAAAC	216
TRPV2	XM_546641.6	TGAACTGCTCTTCCTGGTCC	GCAAGCCGCGTGTGTAATAG	143
TRPV4	NM_001127315.1	CCTGTATGAGTCCTCCGTGG	CTCTGTGGCTGCTTCTCGAT	135
TRPM2	XM_022413206.1	GGACAACGCCTGGATTGAGA	GTGGTTGGCATACAGTGGGA	160
TRPM5	XM_022405530.1	ATGGCAAGTTTGTGAAGGTGC	GACTTCATGGCAAACGACCG	136
TRPM7	XM_535475.6	TCAGCAACTCGTCAGGTGTTT	AAGCATCCGTTGGACTCTGT	167
RPS-19	Xm_533657	CCTTCCTCAAAAA/GTCTGGG	GTTCTCATCGTAGGGAGCAAG	95

## Data Availability

The data presented in this study are available in the article and Appendix A.

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
