# Peer review of "Functional Expression of TRPV1 Ion Channel in the Canine Peripheral Blood Mononuclear Cells"

_ijms, 2021, doi:10.3390/ijms22063177_

Round 1

Reviewer 1 Report

General comments:

Interesting article looking at the expression and function of TRPV1 in canine peripheral blood mononuclear cells. 

In the introduction, the authors state that there are conflicting results regarding TRP channels and immune cells. However, they do not state the nature of these conflicting results. They address some of this conflict in the discussion. It needs to be stated clearly in the introduction as it sets the stage for understanding the motivation behind these experiments. In addition, the relevance of these conflicting results to this article, how do these experiment attempt to address the conflict is needed. There are other aspects in the discussion that would benefit from being in the introduction including the significantly lower expected capsaicin affinity for canine TRPV1 compared to the more commonly studied mouse, rat and human. 

For the Results, these PBMCs appear to express some variety of TRP channels. From Figure 3, the authors state that TRPV1, TRPV4 and TRPM5 have lower mRNA levels than TRPV2, TRPM2 and TRPM7. The logic for following these TRP channels over some of the others is missing. The supplemental figure with the gel of the qPCR results is never references in the main text. The gel also appears to contradict the quantitation in Figure 3a. The TRPV1 appears as strong as the TRPV2 band. This discrepancy needs to be addressed. 

For Figure 3b, I am very concerned about the appropriateness of this antibody. Several scenarios are suggested in the discussion for why multiple bands would be observed and certainly glycosylation could be a factor though I do not agree with the suggestion that the higher bands are multimers if denaturing conditions were used. From the methods, I am unclear whether they were or not. Another concern is that the highest density is coming from a band much lower than the expected molecular weight. In addition, finding a transmembrane protein in the cytosol is very disturbing. Again, the authors do attempt to address this issue by referencing other articles that also claim to observe cytosolic TRPV1. However, given the multiple unexplained bands being picked up by this antibody, I suspect the results may be an artifact of the antibody. Some sort of confirmation that the antibody is truly picking up TRPV1 is needed. Perhaps an siRNA of TRPV1 as a control. 

The patch clamp data in figure 4 nicely shows a capsaicin specific response that is reversed by capsazepine. The currents are very small though that was expected according to references given in the discussion. The results would have benefited from this information to give the reader context especially if they are expecting the currents that are seen from mouse and human TRPV1. Only one concentration of capsaicin is used however. At least in other TRPV1s, there’s a nice potentiation of current with higher capsaicin concentrations where the voltage dependence shifts to the left. This experiment would be a good confirmation that this canine TRPV1 is behaving as expected even with a lower affinity for capsaicin. In addition, dog TRPV1 has been characterized in a heterologous system. How do the author’s data compare? Is the affinity in the same range? And the voltage dependence? These comparisons would help support the characterization that TRPV1 is indeed the channel responsible in these cells. 

For the temperature dependent experiments, the authors changed to fluo-4 system to monitor calcium concentration. What is causing the spike in fluorescence at each temperature change before settling into a steady state? The methods lists a plate reader for these experiments with a built in incubator (according to the website) which is what I assume is inducing the changes in temperature. But the spikes in fluorescence are not discussed. I am also concerned that the only significant change in fluorescence was seen at 42C. This is fairly atypical of TRPV1 channels and brings into question whether these currents are truly from TRPV1.

Capsaicin responses are further characterized with the fluo-4 system.  In figure 6, the authors show a very slow rise in calcium that is dependent on capsaicin concentration. The dependence on capsaicin is clear, however the kinetics are much slower than is expected for calcium influx through a channel. The subsequent experiments with preincubation of capsazepine furthers my concern because the capsazepine does not fully inhibit the calcium influx. This indicates that some of the calcium is not coming from the purported TRPV1 channels however what could be the cause is unclear. 

The discussion is informative but very long and hard to follow in several places, particularly for how the information being discussed is relevant the data presented in the article. As mentioned above, several pieces of information only mentioned in the discussion would do better in the introduction and would help pair down the discussion. Streamlining the discussion further would also make it more clear what contributions the authors are bringing to the field. 

Language issues:

Define PBMC in line 14.

Piezo, line 43

Polycystin, line 97

Scale bar, line 105

In order to, line 138

Author Response

Answer to Reviewer 1:

Thank you for your time and effort that you have dedicated to providing us with valuable feedback. We appreciate all the insightful comments we received and incorporated suggested changes to improve the manuscript. We have highlighted the changes within the manuscript.

Comment 1:
“In the introduction, the authors state that there are conflicting results regarding TRP channels and immune cells. However, they do not state the nature of these conflicting results. They address some of this conflict in the discussion. It needs to be stated clearly in the introduction as it sets the stage for understanding the motivation behind these experiments. In addition, the relevance of these conflicting results to this article, how do these experiment attempt to address the conflict is needed. There are other aspects in the discussion that would benefit from being in the introduction including the significantly lower expected capsaicin affinity for canine TRPV1 compared to the more commonly studied mouse, rat and human.”
Response 1:
We added the paragraph in the introduction about conflicting results especially associated with TRPV1 expression and different doses used in studies on nerve cells and non-excitable cells such as immune cells, lower capsaicin sensitivity of canine TRPV1 due to lack of conserved phosphorylation site and we underlined that there is no current data regarding TRPV1 in the canine immune cells while a dog is recently gaining more attention as a model for human pathophysiology for instance breast cancer (lines 89-105 and 114 -116).

Comment 2:
“For the Results, these PBMCs appear to express some variety of TRP channels. From Figure 3, the authors state that TRPV1, TRPV4 and TRPM5 have lower mRNA levels than TRPV2, TRPM2 and TRPM7. The logic for following these TRP channels over some of the others is missing”.
Response 2:
Yes, we agree with this and we have incorporated short explanation in the manuscript (lines 376 -379). The putative role of other TRP ion channels in the immune cells is also of great interest to us and we want to address their role in future studies.

Comment 3:
The supplemental figure with the gel of the qPCR results is never references in the main text. The gel also appears to contradict the quantitation in Figure 3a. The TRPV1 appears as strong as the TRPV2 band. This discrepancy needs to be addressed.”
Response 3:
The figure is mentioned in line 595 as Figure S1.
We did gel electrophoresis just to validate primers that were designed de novo (we change the name in the figure caption to avoid confusion, line: 701). We did not use the gel for quantification since it refers to the end of the reaction and thus might be prone to stochastic effects and unreliable quantification. Our quantification was based on the Ct values that are based on the early exponential phase. The differences between Ct values and gel electrophoresis were also addressed by Lee et al. (2006) who underlined that gel electrophoresis of end-point PCR is not always correct for quantifying the qPCR products.

Comment 4:
“For Figure 3b, I am very concerned about the appropriateness of this antibody. Several scenarios are suggested in the discussion for why multiple bands would be observed and certainly glycosylation could be a factor though I do not agree with the suggestion that the higher bands are multimers if denaturing conditions were used. From the methods, I am unclear whether they were or not. Another concern is that the highest density is coming from a band much lower than the expected molecular weight. In addition, finding a transmembrane protein in the cytosol is very disturbing. Again, the authors do attempt to address this issue by referencing other articles that also claim to observe cytosolic TRPV1. However, given the multiple unexplained bands being picked up by this antibody, I suspect the results may be an artifact of the antibody. Some sort of confirmation that the antibody is truly picking up TRPV1 is needed. Perhaps an siRNA of TRPV1 as a control”
Response 4:
Thank you for pointing this out. We completely agree with your point. We have encountered several problems with finding good TRP antibodies. In several publications or ion channel meetings (including personal conversations) the poor quality of commercially available TRP antibodies was also mentioned (Meissner et al., 2011; Virck et al., 2019; Goswami & Islam, 2010). For this reason, we were not able to check the protein expression of other TRP channels, for instance, TRPV2. We were also looking for antibodies that recognize canine sequences where most of the TRP antibodies are raised against human or mouse/rat sequences. This is also the reason why we consider changing the model for human PBMC in future studies. Furthermore, we used polyclonal antibodies, which might be the reason why multiple bands are recognized. We underlined this information in the manuscript (lines: 393 - 396). According to manufacturers, the antibody recognizes human, mouse, and rat TRPV1. That is why we used human TRPV1 lysate to detect the correct TRPV1-immunoreactive band. Also, the antibody was already used in at least two other publications about TRPV1 what we found convincing.
Regarding the denaturing conditions. Yes, we used the denaturing conditions (SDS + reducing agent (based on the B-mercaptoethanol) + 10 minutes at 70°C). We should not have observed any oligomers. However, in the case of membrane proteins (or proteins that have hydrophobic patches), they can preserve multimeric forms (and/or non-covalent interaction) or might show abnormal migration on SDS-PAGE (Rath et al., 2009; Shelake et al., 2017).
Yes, we agree that using siRNA as positive control is a great idea and the primary research project assumed TRPV1 gene silencing. Nevertheless, primary PBMC cells are considered to be cells hard-to transfect and standard protocols (for instance using Lipofectamine) are not the best options (very low yield and greatly reduced viability). To our knowledge, the best way to genetically modify these cells (and especially T cells) is viral transfection, which we plan to use in the future.

Comment 5:
“The currents are very small though that was expected according to references given in the discussion. The results would have benefited from this information to give the reader context especially if they are expecting the currents that are seen from mouse and human TRPV1. Only one concentration of capsaicin is used however. At least in other TRPV1s, there’s a nice potentiation of current with higher capsaicin concentrations where the voltage dependence shifts to the left. This experiment would be a good confirmation that this canine TRPV1 is behaving as expected even with a lower affinity for capsaicin. In addition, dog TRPV1 has been characterized in a heterologous system. How do the author’s data compare? Is the affinity in the same range? And the voltage dependence? These comparisons would help support the characterization that TRPV1 is indeed the channel responsible in these cells.”

Response 5:
We have incorporated suggested information in the results section (lines: 238-241). We used one capsaicin concentration (16 μM) based on the protocol for CD4+ cells described by Bertin et al. (2014). Our main goal was to determine whether canine PBMC are responsive to capsaicin and whether it is TRPV1-dependent based on antagonist treatment. We did not address affinity range or voltage dependence, which indeed is a drawback of the study. We incorporated the paper by Phelps et al. (2005) which is the first study that characterizes canine TRPV1 in a heterologous system in the introduction and we mentioned the results obtained by the authors in the discussion (lines: 477-480).

Comment 6:
“For the temperature dependent experiments, the authors changed to fluo-4 system to monitor calcium concentration. What is causing the spike in fluorescence at each temperature change before settling into a steady state? The methods lists a plate reader for these experiments with a built in incubator (according to the website) which is what I assume is inducing the changes in temperature. But the spikes in fluorescence are not discussed. I am also concerned that the only significant change in fluorescence was seen at 42C. This is fairly atypical of TRPV1 channels and brings into question whether these currents are truly from TRPV1.”
Response 6:
Yes, for Fluo-4 assay we used a plate reader that has an incubator for temperature control. We altered the manuscript regarding spikes in fluorescence after incubation periods and we addressed the significant change in fluorescence as seen at 42°C (lines: 275-276 in the results and 452-459 in the discussion section). We assume it is related to ion channel activity. We had no-cell control on the same plate to monitor if temperature and/or time have overall effect on the Fluo-4 alone - we did not find the same pattern in these wells in comparison to well with cells.

Comment 7:
“Capsaicin responses are further characterized with the fluo-4 system. In figure 6, the authors show a very slow rise in calcium that is dependent on capsaicin concentration. The dependence on capsaicin is clear, however the kinetics are much slower than is expected for calcium influx through a channel. The subsequent experiments with preincubation of capsazepine furthers my concern because the capsazepine does not fully inhibit the calcium influx. This indicates that some of the calcium is not coming from the purported TRPV1 channels however what could be the cause is unclear”
Response 7:
Yes, we agree, those observed kinetics of CAP responses were slower than what might be expected. We addressed this in the discussion section (lines 493-498).
Regarding the results with capsazepine (CPZ). We also found it disturbing that CPZ did not inhibit fully CAP-induced calcium influx. We addressed, however, this problem in the discussion section (lines: 499-515) that the results might depend on the agonist: antagonist concentration ratio and that CPZ potentially can activate TRPA1. We cannot exclude the presence of heteroteramers of V1/A1 that have distinct properties, but this needs further studies, especially on the native cells.

Comment 8:
“The discussion is informative but very long and hard to follow in several places, particularly for how the information being discussed is relevant the data presented in the article. As mentioned above, several pieces of information only mentioned in the discussion would do better in the introduction and would help pair down the discussion. Streamlining the discussion further would also make it more clear what contributions the authors are bringing to the field.”
Response : 8
Yes, it is true and partially it is due to extensive data on the TRPV1 that we wanted to cover. We made some corrections (part of the information we transfer to the introduction section as You suggested).

Comment 9:
Language issues:
Define PBMC in line 14.
Piezo, line 43
Polycystin, line 97
Scale bar, line 105
In order to, line 138
Response : 9
We apologize for the spelling errors. Thank You for pointing them out, we have made the corrections.

Reviewer 2 Report

In this manuscript, which show some characteristics of a review paper, authors present evidence for functional expression of TRPV1 receptor channels in the plasma membrane of canine peripheral blood mononuclear cells using quantitative PCR, Western blotting, whole cell patch clamp recordings and calcium imaging. The TRPV1 agonist capsaicin at micromolar concentrations and temperature steps were used to stimulate the channel and the partial TRPV1 antagonists capsazepine was used to attenuate responses.

The paper is written in an elegant way and without linguistic errors. The results start with a phylogenetic overview of the family of TRP ion channels, which seems more like a review than an original investigation but serves as a smooth transition to authors‘ own results. If editors accept this form of the manuscript, it may be welcome.

My only concern is that rather high (i.w., micromolar) doses of capsaicin have been used to cause the effects indicating the presence of TRPV1. These concentrations may cause unspecific effects in light of the fact that excitable cells respond to capsaicin at nanomolar doses and higher doses are regarded as unspecific. Although this point has been addressed in the discussion with one sentence, it should be discussed in more depth.      

Author Response

Answer to Reviewer 2:

Thank you for your time and effort that you have dedicated to provide us with valuable feedback. We appreciate all insightful comments we received and incorporated suggested changes to improve the manuscript. We have highlighted the changes within the manuscript.

Comment 1
“My only concern is that rather high (i.w., micromolar) doses of capsaicin have been used to cause the effects indicating the presence of TRPV1. These concentrations may cause unspecific effects in light of the fact that excitable cells respond to capsaicin at nanomolar doses and higher doses are regarded as unspecific. Although this point has been addressed in the discussion with one sentence, it should be discussed in more depth.”

Response 1:
Yes, we agree with your point. We did previously extensive literature review and we compare the doses of CAP used in the studies with the immune cells. In most of the studies, we found that concentrations of CAP are greater than 10 μM reaching even 300 μM. We agree that at that high concentration unspecific phenomenon might occur. For this reason, we decided to stay with 16 μM (based on the protocol for CD4+ cell from Bertin et al., 2014) for the electrophysiological studies and the lowest (30 μM) dose of CAP where we were able to detect the changes for calcium assay. We added the paragraph about the slower kinetics of CAP response in PBMC (lines: 493-498). We postulate, that immune cells (as non-excitable cells) have distinct physiology from nerve cells or transfected HEK cells where particular ion channels are overexpressed. Low expression of TRPV1 in immune cells can be associated with lower responsiveness to CAP (lines: 470-498). Furthermore, we suggested that multiple aspects can influence sensitivity to CAP starting from species-dependent differences to cellular events such as phosphorylation, presence of splice variants, or heterotetramers formation that exhibit distinct biophysical properties (lines: 420-446).

We adjusted the manuscript and we added some additional information. The information about CAP concentration range used in the studies on immune cells vs nerve cells is added in the introduction (lines: 93-98). We also addressed the possibility of the lower sensitivity of the canine TRPV to capsaicin due to lack of conserved phosphorylation site (lines: 101-105).